# Impact of Seaweed Liquid Extract Biostimulant on Growth, Yield, and Chemical Composition of Cucumber (*Cucumis sativus*)

**Shimaa M. Hassan** [1], **Mohamed Ashour** [2,*], **Nobumitsu Sakai** [3], **Lixin Zhang** [4], **Hesham A. Hassanien** [5,6], **Ahmed Gaber** [7] and **Gamal A.G. Ammarr** [8,*]

1   Department of Vegetable crops, Faculty of Agriculture (El-Shatby), Alexandria University, Alexandria 21545, Egypt; Shaymaa.hassan@alexu.edu.eg
2   National Institute of Oceanography and Fisheries, NIOF, Cairo 11516, Egypt
3   The Niva Labs13171 Telfair Ave, Suite B, Sylmar, CA 91342, USA; umkyotosakai@gmail.com
4   Researcher, College of Life Sciences, Northwest A&F University, Yangling 712100, China; zhanglixin@nwsuaf.edu.cn
5   Animal and fish Production Department, College of Agricultural and Food Sciences, King Faisal University, P.O. Box 420, Al-Ahsa 31982, Saudi Arabia; helsanwey@kfu.edu.sa
6   Animal Production Department, Faculty of Agriculture, Cairo University, Gamma St., Giza 12613, Egypt
7   Department of Biology, College of Science, Taif University, P.O. Box 11099, Taif 21944, Saudi Arabia; a.gaber@tu.edu.sa
8   Biotechnology Unit., Plant Production Department, Arid Lands Cultivation Research Institute, City of Scientific Research and Technological Applications, Alexandria 2193, Egypt
*   Correspondence: egyptmicroalgae@gmail.com (M.A.); gammar@srtacity.sci.eg (G.A.G.A)

**Abstract:** Seaweed extract biostimulants are among the best modern sustainable biological plant growth promoters. They have been proven to eliminate plant diseases and abiotic stresses, leading to maximizing yields. Additionally, they have been listed as environmentally friendly biofertilizers. The focus of the present research is the use of a commercial seaweed biostimulant as an eco-friendly product (formally named True Algae Max (TAM). During the 2017 and 2018 seasons, five treatments of various NPK:TAM ratios were applied via regular fertigation, namely a conventional treatment of 100% NPK ($C_0$) alongside combinations of 25%, 50%, 75%, and 100% ($C_{25}$, $C_{50}$, $C_{75}$, and $C_{100}$) of TAM, to evaluate the effectiveness of its bioactive compounds on enhancing growth, yield, and NPK content of cucumber (*Cucumis sativus*) under greenhouse conditions. TAM is rich in phytochemical compounds, such as milbemycin oxime, rhodopin, nonadecane, and 5-silaspiro [4.4]nona-1,3,6,8-tetraene,3,8-bis(diethylboryl)-2,7-diethyl-1,4,6,9-tetraphenyl-. Promising measured parameter outcomes showed the potentiality of applying TAM with and without mixes of ordinary NPK application. TAM could increase cucumber yield due to improving chemical and physical features related to immunity, productivity, and stress defense. In conclusion, it is better to avoid applying mineral fertilizers, considering also that the organic agricultural and welfare sectors could shortly depend on such biotechnological tools and use them to fulfill global food demands for improved sustainability.

**Keywords:** TAM; FAMEs; seaweed biostimulants; *Ulva lactuca*; *Jania rubens*; *Pterocladia capillacea*

## 1. Introduction

The extensive utilization of chemical fertilizer causes important pollution impacts on societies, with harmful socio-economic and environmental effects. Recently, greenhouse gas production has been increasing, causing global warming and groundwater pollution, which are mainly linked to climatic changes and directly affect the future of agricultural business [1]. Seeking safe, environmentally friendly, and brighter sustainability represents a high priority facing modern plant production [2]. Novel alternative approaches, such as applying plant bioeffectors formally known as plant biostimulants, have become more

widely adopted and have become commonly applied in a variety of agricultural practices, providing a number of benefits in stimulating plant growth and protecting against stresses [3].

Seaweed extracts are amongst the most commonly utilized important sustainable biostimulants [4]. The employment of seaweed extracts as biostimulants in agricultural practices is very ancient, having been in use since early plant breeding [3]. Based on their nutritional values, algal cells (either microalgae or seaweeds) represent a treasury of sources of dyes, proteins, fats, polyunsaturated fatty acids (PUFA), polysaccharides, minerals, and antioxidants, as well as plenty of biological components [5–9]. In addition, seaweeds have important environmental functions, being one of the most important components of marine ecosystems [10]. On the other hand, seaweed extracts could be used in various biological processes in industry [11]. Moreover, the use of algal cells and/or their extracts as aquaculture feeds enhances aquatic growth and immunity [12–14]. Seaweeds are qualified to be used as biofertilizers, not only because they have a biological impact, but also because of their biocompatibility as they share common biological compounds with plants. This major advantage has put seaweeds at the top of the plant biostimulant list and facilitated many plant treatment processes, mainly for serving and facilitating organic and sustainable agriculture [15]. The positive impacts of seaweed extracts as plant biostimulants have been reported in the literature. Some of the most frequently used extracts are reported to be *Pterocladia capillacea* [4], *Ascophyllum nodosum* [16], *Ecklonia maxima*, *Sargassum* spp. [17], *Ulva lactuca*, *Caulerpa sertularioides*, *Padina gymnospora*, *Sargassum liebmannii* [18], *U. lactuca*, *Laminaria* spp., *P. gymnospora*, *Durvillaea potatumum*, *C. sertularioides*, *S. johnstonii*, and *S. liebmannii* [19]. Often, some undefined biologically effective compounds in seaweed extracts can cause plants to produce their phytohormones through internal metabolic pathways [4].

Despite the availability of different seaweed extract supplementations for plant techniques, foliar spray administration in modern agriculture has been sufficiently and broadly used to increase the yield of many commercial crops, with very promising results. As reported previously, seaweed liquid fertilizers could increase chlorophyll content [20], increase total yield [4], and improve the root system [21]. In conjunction with its rapid and easy handling process, the applicability of seaweed extract foliar spraying has been studied [22], with a focus on stimulating growth and increasing the productivity of some important vegetable crops, such as cucumber [23–26].

Biologically, seaweed extracts are dissimilar from chemical fertilizers. They are characterized by being biodegradable and harmless, making them environmentally friendly substances with no chemical residues and/or hazards [4]. Consequently, the screening of innate aquatic species must be considered in order to achieve a successful commercial and biotechnological potentiality [12]. The Egyptian coast, including the Mediterranean and the Red Sea coasts, has a wide range of wild seaweed, harvested throughout the year [4,8]. On the Egyptian Mediterranean coast, especially near Alexandria, red algae *Pterocladia capillacea* and *Jania rubens* and green alga *Ulva lactuca* are the most dominant native seaweed species. The unique and exceptional nature and structure of algal extracts and derivatives have important biotechnological potentials for disease resistance [27]. Moreover, they supply crops with nutrients, encourage the production of superior biomass, and activate the natural ability of plants to cope with environmental stresses. Because of their capability to contain a miscellany of biologically active molecules, their ability to have a beneficial influence on plants has been substantiated. With very promising stable steps, seaweed extracts are highly recommended for use in assisting organic agriculture [4]. There are a lot of available, newly emerging seaweed extract products in the market with a quickly accelerating expansion in production and competitive market share with chemical fertilizers and pesticides; therefore, as a deduction, the objective of the current research is mainly to study the comparableness of the performance of a commercial seaweed extract (True Algae Max (TAM)) as a foliar spray application on cucumber *Cucumis sativus* in greenhouse

with regard to conventional NPK foliar spray fertilizing routines, as a step on the way to adopting the usage of biostimulants rather than chemical ones.

## 2. Materials and Methods

### 2.1. Seaweed Liquid Extract Methods

TAM is a commercial seaweed liquid extract, submitted as a patent [11]. The preparation of TAM was as described by [13]. In detail, three seaweed species, *Ulva lactuca* (Chlorophyceae), *Jania rubens*, and *Pterocladia capillacea* (Rhodophyceae), were employed in preparing and producing TAM. The selected species were collected in the 2016 summer season from the rocky site (31°16′16.0″ N, 30°10′28.0″ E) of Abu-Qir Bay, the Mediterranean Coast of Alexandria, Egypt. After being harvested, the epiphytic and waste materials were removed, then samples were washed, air-dried, powdered, and finally kept in plastic bags at room temperature for further analysis. Phytochemical, physical, chemical, and biochemical analyses of crude TAM were conducted and estimated as described by Ashour et al. [13], as presented in Figure 1 and Tables 1 and 2.

**Table 1.** Physical, chemical, and biochemical analyses of TAM.

| Item | Value |
|:---:|:---:|
| Physical analyses | |
| Color | Dark brown |
| Odor | Seaweed |
| Density | 1.20 |
| pH | 9–9.5 |
| Biochemical analyses (% DM) | |
| Total polysaccharides | 15 |
| Total organic matter | 8.2 |
| Total dissolved solids | 2.6 |
| Chemical analyses | |
| Major elements (%) | |
| Potassium | 15 |
| Phosphorus | 2.4 |
| Total nitrogen | 0.14 |
| Minor elements (ppm) | |
| Copper | 0.39 |
| Iron | 16.18 |
| Magnesium | 19.72 |
| Zinc | 1.19 |
| Manganese | 3.72 |
| Heavy metals (ppm) | |
| Cadmium | 0.00 |
| Chromium | 0.00 |
| Lead | 0.00 |
| Nickel | 0.00 |
| Arsenic | 0.55 |

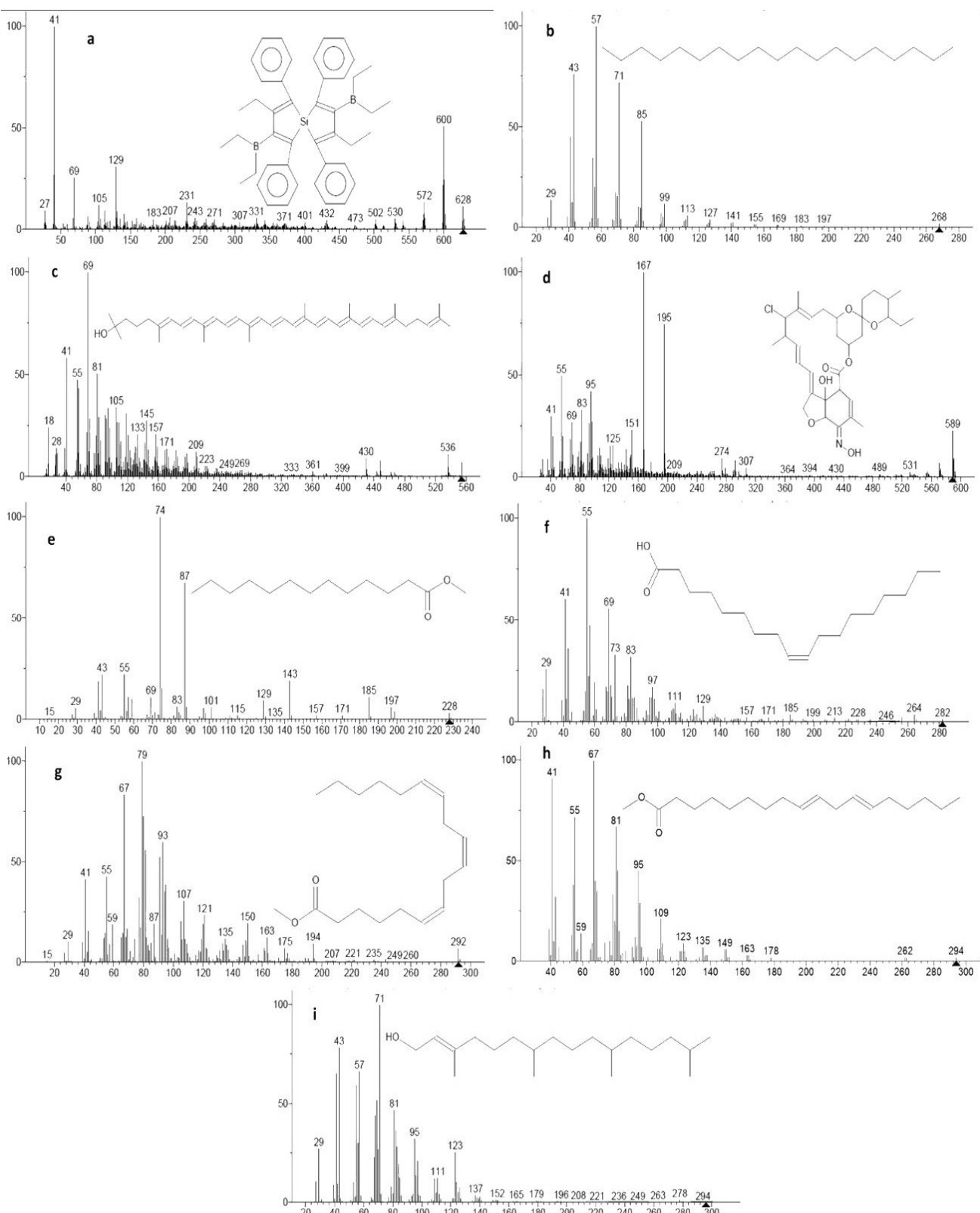

**Figure 1.** Mass spectra and retention times of nine phytochemical compounds in True Algae Max (TAM) identified with the National Institute of Standards and Technology (NIST) library [17]. (**a**) 5-Silaspiro[4.4]nona-1,3,6,8-tetraene,3,8-bis(diethylboryl)-2,7-diethyl-1,4,6,9-tetraphenyl- (8.99 min); (**b**) nonadecane (16.31 min); (**c**) rhodopin (19.45 min); (**d**) milbemycin B (20.07 min); (**e**) tridecanoic acid methyl ester (20.90 min); (**f**) oleic acid (21.63 min); (**g**) γ-linolenic acid methyl ester (23.74 min); (**h**) 9,12-octadecadienoic acid, methyl ester, (E,E)- (24.02 min); and (**i**) phytol (24.37 min).

## 2.2. Cucumber Cucumis sativus Methods

2.2.1. Soil Analysis

The experiment was carried out during 2017 and 2018 seasons under greenhouse conditions (about 300 $m^2$) at Abis Experimental Farm Station (31.2001° N and 29.9187° E), the Faculty of Agriculture, Alexandria University, Egypt. Before each experiment, in both the 2017 and 2018 seasons, soil samples were collected from greenhouses at a 15–30 cm depth and analyzed at the Central Laboratory, Faculty of Agriculture, Alexandria University. The physical and chemical properties of the soil were determined according to Page et al. [28], as presented in Table 3.

**Table 2.** Phytochemical compounds of TAM and related biological activities according to the literature.

| RT (min) | Compound Name | Formula | Molecular Weight | Nature | Biological Properties | Literatures |
|---|---|---|---|---|---|---|
| 8.99 | 5-Silaspiro[4.4]nona-1,3,6,8-tetraene,3,8-bis(diethylboryl)-2,7-diethyl-1,4,6,9-tetraphenyl- | $C_{44}H_{50}B_2Si$ | 628.39 | Silicon–boron compound | Growth and immunity enhancer; antifungal for plant pathogen | [13,29–31] |
| 16.31 | Nonadecane | $C_{19}H_{40}$ | 268.31 | Alkane | Antioxidant; anticancer; antimicrobial; anti-inflammatory | [32–36] |
| 19.45 | Rhodopin | $C_{40}H_{58}O$ | 554.45 | Carotenoid | Growth enhancer; antioxidant | [13,37,38] |
| 20.07 | Milbemycin B, 5-demethoxy-5-one-6,28-anhydro-25-ethyl-4-methyl-13-chloro-oxime | $C_{32}H_{44}ClNO_7$ | 589.28 | Macrocyclic lactones | Immunity enhancer; antiparasitic; antihelminthic; insecticidal | [13,39–41] |
| 20.90 | Tridecanoic acid, methyl ester | $C_{14}H_{28}O_2$ | 228.21 | Fatty acid methyl esters (FAMEs) | Antioxidant; surfactants; herbicidal; antimicrobial | [13,42–44] |
| 21.63 | Oleic acid | $C_{18}H_{34}O_2$ | 282.26 | Fatty acid | Immunity enhancer; enhancing insulin production; anti-inflammatory | [13,45–48] |
| 23.74 | γ-Linolenic acid, methyl ester | $C_{19}H_{32}O_2$ | 292.24 | FAMEs | Antioxidant; surfactants; herbicidal; antimicrobial | [13,42–44] |
| 24.02 | 9,12-Octadecadienoic acid, methyl ester, (E,E)- | $C_{19}H_{34}O_2$ | 294.26 | FAMEs | Antioxidant; surfactants; herbicidal; antimicrobial | [13,42–44] |
| 24.37 | Phytol | $C_{20}H_{40}O$ | 296.31 | Diterpene alcohol | Antioxidant; antinociceptive | [13,49,50] |

2.2.2. Experimental Design

The current experiment was performed to evaluate the impact of applying diverse levels of TAM foliar spray as a growth biostimulant and NPK mineral fertilizer, alone or in combination, on the growth and fruit yield of cucumber *Cucumis sativus* (Hesham $F_1$) grown under greenhouse conditions with three replicates. Each replicate was conducted in five ridges of 100 $m^2$ (5 rows × 1 m width × 20 m long). The fertilizer treatments were $C_0$: 100% conventional NPK mineral fertilizer as control, $C_{25}$: 25% TAM (1.25 mL of crude extract/liter) + 75% NPK mineral fertilizer, $C_{50}$: 50% TAM (2.5 mL of crude extract/liter) + 50% NPK mineral fertilizer, $C_{75}$: 75% TAM (3.75 mL of crude extract/liter) + 25% NPK mineral fertilizer, and $C_{100}$: 100% TAM (5 mL of crude extract/liter).

The experimental layout was a randomized complete block design (RCBD). Each treatment had three replicates, and each replicate had five plots, where each plot was 20 $m^2$ (1 m wide and 20 m long). Seeds were sown on the 20th of January for both the 2017

and 2018 growing seasons. During the entire growing season, TAM was added as a foliar spray two times weekly. The conventional NPK control treatment was added once weekly through a drip irrigation system in equal doses, starting one week after transplanting. After transplanting, ammonium nitrate (33% N, as a source of nitrogen) was added at the rate of 10 kg/100 $m^2$/12 weeks; phosphoric acid (61.5% $P_2O_5$, as a source of phosphorus) was injected at the rate of 1.85 L/100 $m^2$/12 weeks, and potassium sulfate (48% $K_2O$, as a source of potassium) was added at the rate of 8 kg/100 $m^2$/12 weeks. A schedule of NPK fertigation applied in the current study for cucumber grown in loamy soil under greenhouse [51,52] is presented in Table 4.

**Table 3.** Soil physical and chemical properties of the experimental sites during the two growing seasons of 2017 and 2018.

| Properties | Seasons | |
|---|---|---|
| | Winter 2017 | Winter 2018 |
| Physical properties | | |
| Sand% | 43.3 | 42.8 |
| Silt% | 25.5 | 23.5 |
| Clay% | 31.2 | 33.7 |
| Soil texture | Loamy | Loamy |
| Chemical properties | | |
| pH | 8.45 | 8.88 |
| E.C. (dS $m^{-1}$) | 3.01 | 3.00 |
| Soluble cations (m.eq/L) | | |
| $Ca^+$ | 2.13 | 2.02 |
| $Mg^{++}$ | 2.03 | 1.93 |
| $Na^+$ | 2.51 | 2.43 |
| $K^+$ | 0.41 | 0.38 |
| Soluble anions (m.eq/L) | | |
| $CO_3$ | Zero | Zero |
| $HCO_3$ | 1.35 | 1.20 |
| $Cl^-$ | 2.00 | 1.90 |
| $SO^-$ | 3.20 | 3.11 |
| Total nitrogen (%) | 0.19 | 0.15 |
| Phosphorus (ppm) | 0.41 | 0.44 |

**Table 4.** Schedule of fertigation system (100% conventional N, P, K mineral fertilizer as a control treatment) applied for cucumber grown in loamy soil during the two growing seasons of 2017 and 2018 under a greenhouse.

| NPK Rate | Week after Transplanting | Ammonium Nitrate (g 100 $m^{-2}$) | Phosphoric Acid (cm$^3$ 100 $m^{-2}$) | Potassium Sulfate (g 100 $m^{-2}$) |
|---|---|---|---|---|
| 2% | 2 | 200 | 37 | 160 |
| 4% | 3 | 400 | 74 | 320 |
| 6% | 4 | 600 | 111 | 480 |
| 8% | 5 | 800 | 148 | 640 |
| 12% | 6 | 1200 | 222 | 960 |
| 12% | 7 | 1200 | 222 | 960 |
| 12% | 8 | 1200 | 222 | 960 |
| 12% | 9 | 1200 | 222 | 960 |
| 8% | 10 | 800 | 148 | 640 |
| 8% | 11 | 800 | 148 | 640 |
| 8% | 12 | 800 | 148 | 640 |
| 8% | 13 | 800 | 148 | 640 |

### 2.2.3. Tested Parameters

*Cucumis sativus* Growth Parameters

One hundred days after sowing, ten plants from each experimental unit were randomly chosen for the vegetative growth measurable values of plant height (from soil surface to the top of the plant in cm) and number of leaves. Moreover, the average leaf area ($cm^2$) of the sixth leaf from the meristematic tip of ten randomly selected plants was determined by using a "planimeter", and then the average leaf area per plant was calculated.

*Cucumis sativus* Yield and Fruit Parameters

Total yield was evaluated through the weight of all harvested fruits per pot within the entire harvesting season (kg), then transformed and calculated as total yield per square meter ($kg\ m^{-2}$). Fruit length (L) was measured for of ten randomly selected fruits, and the average fruit length in centimeters was calculated. Moreover, fruit diameter (D) was measured by a vernier caliper for randomly chosen ten fruits, and then the average head diameter in centimeters was calculated.

Chlorophyll Content of Leaves

Freshly picked leaves from ten plants of each treatment were used to determine the total chlorophyll content of the leaves ($mg.100\ g\ m^{-1}$ f.w.). The total chlorophyll content ($\mu E = \mu Mol\ m^{-2}\ s^{-1}$) of fresh leaves was determined by using a chlorophyll fluorometer (model OPTI-SCIENCES OS-30; Opti-sciences, Inc., Hudson, NH, USA).

Dry Matter Content of Leaves and Fruits

Random samples of 50 g fresh weight from leaves and fruits were used separately, then dried at 70 °C until the weight became constant. After that, the percentages of the dry matter contents of the leaves and fruits (%) were calculated.

Mineral Contents of Leaves and Fruits

Samples collected from each plot were cleaned using distilled water washing, then oven-dried at 70 °C to a constant weight. Total N and P contents were determined calorimetrically, using a spectrophotometer, at 662 and 650 nm [53], while K was determined by atomic absorption spectrometry [54].

### 2.3. Statistical Analysis

Analysis of variance (ANOVA) was used to distinguish differences among means, which were considered significant at a $p < 0.05$ degree of possibility, obtaining the least significant difference (LSD) to resolve the differences among means of replication according to Duncan by SPSS. The results of anticancer activity were used to calculate the $IC_{50}$ values of each extract, using probit analysis and the SPSS computer program (SPSS for Windows, statistical analysis software package, version 9, 1989; SPSS Inc., Chicago, IL, USA).

## 3. Results

### 3.1. Cucumber Cucumis sativus

#### 3.1.1. Growth

The results in Table 5 show the influence of different doses of TAM on the growth characteristics of *C. sativus* under greenhouse conditions during the two growing seasons of 2017 and 2018. The plant height trait was recorded as 193.7 cm in 2017 under treatment with $C_0$, while it reached 209.7 cm in 2018 under treatment with $C_{100}$. Additionally, the number of leaves was significantly affected by the application of TAM and averaged between 27.7 cm (in 2017 under $C_0$ treatment) and 306 cm (under $C_{75}$ treatment in 2018). A similar pattern was observed for the leaf area parameter, where significant differences were influenced by the use of TAM; the leaf area parameter was listed as 306 plant/$cm^{-2}$ in the 2017 season, while it was 275 plant/$cm^{-2}$ in the same season under conventional mineral fertilization with $C_0$. On the one hand, there were no significant observed differences

concerning chlorophyll content measurements, which ranged from 37.9 mg/100 gm f.w. the in 2017 season without any TAM treatments to a slightly higher record (39.4 mg/100 gm f.w.) in the 2018 season when applying $C_{100}$. On the other hand, the effect of TAM on promoting plant growth parameters could also represent significant returns in regard to the dry matter percentage, where under controlled conditions of 100% conventional mineral fertilization ($C_0$), a dry matter percentage of 14.6% was recorded for the 2018 cultivating season, while it reached of 16.6% when applying TAM $C_{75}$ in the same season. For the N content parameter, the measurements revealed that controlled condition resulted in N content of 3.48% in the 2018 season, while it declined to be 2.91% in the same year when using $C_{100}$. However, it was noticed that there were no clear significant differences altered by TAM for the P and K percentages during the two seasons (Table 5).

### 3.1.2. Yield and Chemical Composition

Table 6 shows the influence of different doses of TAM on the fruit characteristics of cucumber *C. sativus* in the greenhouse during the two cultivating seasons, 2017 and 2018. It was recorded that the total yield was powerfully affected by the treatment with $C_{50}$ and $C_{75}$ in the 2017 season, reaching 4.07 kg/m$^2$ for both concentrations, while it reached as low as 2.47 kg/m$^2$ in the same season without treatment. For the dry matter content trait, the measurements showed that the percentage of dry matter ranged from 4.48% under controlled conditions in 2017 up to 5.29% in 2018 under the application of $C_{25}$ and $C_{100}$ TAM, independently. Although the dry matter content had been slightly increased by exploiting different concentrations of TAM, this boost did not appear to show an effective significant difference among treatments (Table 6). Approximately, the same trend was observed from the results of fruit length, which was 14.3 cm in 2017 for untreated plants and reached approximate stability across the two seasons with various TAM concentrations; however, the maximum fruit length was recorded as 15.2 cm in 2017 with $C_{100}$ treatment, while the minimum fruit length was recorded as 13.3 cm in 2018 without any treatments. Importantly, no notable significant differences could be distinguished for the impact of TAM treatments on the NPK elements of fruit quality during the two successive seasons (Table 6).

**Table 5.** The effect of different TAM doses on the growth characteristics of cucumber *C. sativus* and N, P, K content cultured under a greenhouse during the 2017/2018 growing seasons.

| Treatments * | $C_0$ | | $C_{25}$ | | $C_{50}$ | | $C_{75}$ | | $C_{100}$ | |
|---|---|---|---|---|---|---|---|---|---|---|
| Parameters | 2017 | 2018 | 2017 | 2018 | 2017 | 2018 | 2017 | 2018 | 2017 | 2018 |
| Plant height (cm) | 193.7 ± 2.3 [b] | 195.3 ± 2.5 [b] | 199.0 ± 1.0 [a,b] | 197.3 ± 4.0 [b] | 200.0 ± 3.5 [a,b] | 200.3 ± 3.2 [b] | 200.7 ± 3.8 [a,b] | 201 ± 2.6 [b] | 203.3 ± 5.9 [a] | 209.7 ± 1.5 [a] |
| Leaf number | 27.7 ± 1.2 [d] | 28.3 ± 0.6 [c] | 29.7 ± 0.6 [c] | 30.7 ± 0.6b [c] | 30.7 ± 0.6 [b,c] | 32.3 ± 0.6 [a] | 32.3 ± 0.6 [a] | 32.6 ± 0.6 [a] | 31.3 ± 1.2 [a,b] | 31.6 ± 1.1 [a,b] |
| Leaf area (plant/cm$^2$) | 275 ± 5.0 [c] | 288 ± 3.0 [b] | 299 ± 1.5 [b] | 299 ± 2.3 [a] | 301 ± 0.6 [a,b] | 304 ± 4.5 [a] | 306 ± 4.2 [a] | 306 ± 1.5 [a] | 305 ± 2.9 [a,b] | 306 ± 10.4 [a] |
| Chlorophyll (mg.100g f.w.) | 37.9 ± 0.7 [a] | 38.7 ± 0.5 [a] | 38.9 ± 0.4 [a] | 39.0 ± 0.4 [a] | 39.3 ± 0.1 [a] | 39.0 ± 1.5 [a] | 38.8 ± 1.6 [a] | 38.7 ± 0.5 [a] | 38.7 ± 1.6 [a] | 39.4 ± 0.9 [a] |
| Dry matter (%) | 14.8 ± 0.9 [b] | 14.6 ± 1.0 [a] | 15.1 ± 0.2 [a b] | 15.4 ± 0.6 [a b] | 15.7 ± 0.4 [a,b] | 15.9 ± 0.5 [a,b] | 16.4 ± 0.5 [a] | 16.6 ± 0.7 [a] | 15.8 ± 0.9 [a,b] | 16.6 ± 0.5 [a] |
| Leaf nitrogen (%) | 3.25 ± 0.14 [a] | 3.48 ± 0.55 [a] | 2.96 ± 0.28 [a] | 2.99 ± 0.13 [a,b] | 3.21 ± 0.17 [a] | 3.27 ± 0.14 [a b] | 3.01 ± 0.20 [a] | 2.98 ± 0.15 [a,b] | 3.03 ± 0.06 [a] | 2.91 ± 0.17 [b] |
| Leaf phosphorus (%) | 0.51 ± 0.02 [a] | 0.55 ± 0.05 [a] | 0.48 ± 0.07 [a] | 0.47 ± 0.06 [a] | 0.52 ± 0.02 [a] | 0.52 ± 0.03 [a] | 0.48 ± 0.04 [a] | 0.51 ± 0.01 [a] | 0.54 ± 0.01 [a] | 0.50 ± 0.03 [a] |
| Leaf potassium (%) | 2.97 ± 0.09 [a] | 3.02 ± 0.42 [a] | 2.92 ± 0.07 [a] | 2.87 ± 0.12 [a] | 2.97 ± 0.09 [a] | 2.96 ± 0.17 [a] | 2.80 ± 0.20 [a] | 2.96 ± 0.03 [a] | 2.83 ± 0.16 [a] | 2.96 ± 0.09 [a] |

\* Represented data are mean ± SD (*n* = 3). Different superscript letters in each row indicate significant differences (*p* < 0.05). TAM: seaweed liquid extract; $C_0$: 100% NPK mineral fertilizer as control; $C_{25}$: 25% TAM (1.25 cm of crude extract/liter) + 75% NPK mineral fertilizer; $C_{50}$: 50% TAM (2.5 cm of crude extract/liter) + 50% NPK mineral fertilizer; $C_{75}$: 75% TAM (3.75 cm of crude extract/liter) + 25% NPK mineral fertilizer; and $C_{100}$: 100% TAM (5 cm of crude extract/liter).

**Table 6.** The influence of different TAM doses on the fruit characteristics of cucumber *C. sativus* and N, P, K content cultured under a greenhouse during the 2017/2018 growing seasons.

| Treatments * | $C_0$ | | $C_{25}$ | | $C_{50}$ | | $C_{75}$ | | $C_{100}$ | |
|---|---|---|---|---|---|---|---|---|---|---|
| Parameters | 2017 | 2018 | 2017 | 2018 | 2017 | 2018 | 2017 | 2018 | 2017 | 2018 |
| Total yield (kg/m$^2$) | 2.47 ± 0.42 [c] | 2.80 ± 0.10 [c] | 3.40 ± 0.20 [b] | 3.47 ± 0.25 [b] | 4.07 ± 0.12 [a] | 3.93 ± 0.23 [a] | 4.07 ± 0.15 [a] | 4.01 ± 0.02 [a] | 3.93 ± 0.23 [a] | 4.00 ± 0.00 [a] |
| Dry matter (%) | 4.84 ± 0.46 [a] | 4.85 ± 0.25 [b] | 5.11 ± 0.19 [a] | 5.29 ± 0.15 [a] | 5.20 ± 0.18 [a] | 5.15 ± 0.12 [a,b] | 5.17 ± 0.17 [a] | 5.19 ± 0.19 [a] | 5.15 ± 0.03 [a] | 5.29 ± 0.07 [a] |
| Length (Cm) | 14.3 ± 0.86 [a] | 13.3 ± 0.16 [b] | 14.7 ± 0.45 [a] | 14.4 ± 0.69 [a] | 14.9 ± 0.47 [a] | 14.9 ± 0.53 [a] | 14.7 ± 0.53 [a] | 15.1 ± 0.14 [a] | 15.2 ± 0.20 [a] | 15.1 ± 0.21 [a] |
| Diameter (Cm) | 3.06 ± 0.06 [b] | 3.08 ± 0.06 [a] | 3.09 ± 0.01 [b] | 3.09 ± 0.02 [a] | 3.15 ± 0.06 [b] | 3.09 ± 0.13 [a] | 3.26 ± 0.25 [b] | 3.13 ± 0.11 [a] | 3.57 ± 0.16 [a] | 3.11 ± 0.04 [a] |
| Fruit nitrogen (%) | 2.24 ± 0.28 [a] | 2.09 ± 0.08 [a] | 2.00 ± 0.01 [a b] | 1.97 ± 0.07 [a] | 2.05 ± 0.04 [a,b] | 2.06 ± 0.05 [a] | 1.97 ± 0.05 [b] | 2.04 ± 0.05 [a] | 2.03 ± 0.04 [a b] | 2.00 ± 0.08 [a] |
| Fruit phosphorus (%) | 0.41 ± 0.02 [a] | 0.47 ± 0.03 [a] | 0.42 ± 0.03 [a] | 0.43 ± 0.06 [a] | 0.40 ± 0.02 [a] | 0.47 ± 0.06 [a] | 0.43 ± 0.06 [a] | 0.40 ± 0.02 [a] | 0.41 ± 0.02 [a] | 0.42 ± 0.04 [a] |

\* Represented data are mean ± SD (*n* = 3). Different superscript letters in each row indicate significant differences (*p* < 0.05). TAM: seaweed liquid extract; $C_0$: 100% NPK mineral fertilizer as control; $C_{25}$: 25% TAM (1.25 cm of crude extract/liter) + 75% NPK mineral fertilizer; $C_{50}$: 50% TAM (2.5 cm of crude extract/liter) + 50% NPK mineral fertilizer; $C_{75}$: 75% TAM (3.75 cm of crude extract/liter) + 25% NPK mineral fertilizer; and $C_{100}$: 100% TAM (5 cm of crude extract/liter).

## 4. Discussion

The research presented in this paper was mainly conducted to study the effect of foliar application of seaweed commercialized extract (TAM) on the growth, yield, and quality of cucumber (*Cucumis sativus)* in comparison with conventional NPK fertilization. For this purpose, the tested hypothesis assumed that foliar application of TAM would result in an increase in the yield components and fruit quality of cucumber on a physical and/or chemical basis. Table 5 demonstrates that the TAM physical and chemical analyses, regarding its biosafety, show that it is a harmless compound with safe applicability and, remarkably, that it contains relatively no amounts of heavy metals. Appropriate combinations of serial concentrations from NPK and TAM mixes in ratios of 25% ($C_{25}$), 50% ($C_{50}$), 75% ($C_{75}$), and, finally, 100% pure TAM ($C_{100}$), were subsequently prepared to examine the effectiveness of the foliar spray administration versus the untreated control using 100% conventional NPK mineral fertilizer. The results indicated that the growth parameters were effectively influenced by the application of different TAM concentrations in a very promising magnitude (Table 5). With regard to traits such as plant height, leaf number, leaf area, and dry matter percentage, a tangible variation over the two seasons of cultivation could be induced. Comparatively, the doses used across the chlorophyll content and measurements of N, P, and K did not show the same impact of variation between the controlled treatment and the TAM ones. Similarly, Table 6 shows that treatments with different quantities of TAM were capable of positively enhancing fruit quality parameters such as total yield, fruit length, and diameter, with respect to the treatments with an almost equal effect; this similarly followed the same trend observed when administrating mineral fertilization as well as for the N, P, and K trait analyses, which was reflected in dry matter percentage, Our findings were in line with those of Hamed et al. [55], who stated that seaweed extracts could contain nutrients, trace minerals, vitamins, phytohormones, ascorbic acids, and many other bioactive compounds [56].

According to Ashour et al. [13], the physical, chemical, and biochemical analyses of TAM, which were used in the current study as biostimulator foliar sprays for cucumber (Table 2), show that TAM has a dark brown color, a seaweed odor, a density of 1.2, and a pH of 9–9.5. TAM showed large amounts of macronutrients, including K (15%) and P (2.4%), while small amounts of N (0.1%) were reported. Micronutrients such as Cu, Fe, Mg, Zn, and Mn (0.39, 16.18, 19.72, 1.19, and 3.72 ppm, respectively) also existed in small amounts as traces. Moreover, data for heavy metals show that cadmium, chromium, lead, and nickel each presented a concentration of 00.00 ppm, while a concentration of 0.55 ppm was found for arsenic. Biochemical analyses of the total polysaccharides, total organic matter, and total dissolved solids were recorded to be 15%, 8.2%, and 2.6% (percent based on dry matter bases), respectively (Table 2).

Marine aquatic organisms are an important source of bioactive compounds [57,58]. In the current study, concerning the GC-MS analysis [13,59,60], the NIST (National Institute of Standards and Technology) mass spectral library identified nine compounds in TAM® (Figure 1). Most compounds had a long hydrocarbon chain, such as nonadecane (alkane), rhodopin (carotenoid), oleic acid (fatty acid), phytol (diterpene alcohol), milbemycin oxime (macrocyclic lactones), and the fatty acid methyl esters (FAMEs) (tridecanoic acid methyl ester, γ-linolenic acid methyl ester, and 9,12-octadecadienoic acid, methyl ester, (E,E)-). TAM also showed many valuable bioactive phytochemical compounds not only for fish and plant growth enhancement but also for industrial applications, especially for the pharmaceutical and medical industries. The fragmentation patterns of TAM compounds (Figure 1), such as base peak and successive loss of 14 atomic mass units ($CH_2$ unit), matched with previous reports [61]. Interestingly, oleic acids are the most dominant fatty acids in seaweeds [45–48]. Tridecanoic acid methyl ester; γ-linolenic acid methyl ester; and 9,12-octadecadienoic acid, methyl ester, (E,E)- are FAMEs that are environmentally safe, nontoxic, and biodegradable compounds used for biodiesel and antimicrobial activities [42–44]. FAMEs have a high affinity for fatty compounds which are suitable for penetrating the plant cuticle, as well as facilitating the attachment of the sprayed mixture to the leaf surface [44]. Phytol is a

product of chlorophyll metabolism in plants and is abundantly available in nature [49]. It is also involved with tocopherol (vitamin E), phylloquinol (vitamin K), and fatty acid phytyl ester productions. Phytol has demonstrated strong effects as an antioxidant [50]. Rhodopin is also an antioxidant compound [37] found in marine seaweeds [38]. Nonadecane is also a natural alkane hydrocarbon that was recorded in different brown, red, and green seaweeds, collected from the Alexandrian coast. However, nonadecane showed different biological properties such as antimicrobial and antioxidant activities [32–36]. Milbemycin B, 5-demethoxy-5-one-6,28-anhydro-25-ethyl-4-methyl-13-chloro-oxime (milbemycin oxime), was also reported in TAM [13,59,60]. Milbemycins are a group of macrocyclic lactones. Additionally, they have anthelminthic and some insecticidal properties [49]. Milbemycins were originally isolated from *Streptomyces hygroscopicus* subsp. *aureolacrimosus* [40], and milbemycin-producing strains have been widely found all over the world [41]. Interestingly, milbemycin oxime was reported previously in microalga *Chlorella sorokiniana* [50]; however, to the best of our knowledge, there are no reports for detecting the presence of milbemycin oxime in seaweed extracts (such as TAM) prior to that of Ashour et al. [13].

In addition to the existing compounds, another important constituent, namely 5-silaspiro[4.4]nona-1,3,6,8-tetraene,3,8-bis(diethylboryl)-2,7-diethyl-1,4,6,9-tetraphenyl-, was found also in TAM. 5-Silaspiro[4.4]nona-1,3,6,8-tetraene,3,8-bis(diethylboryl)-2,7-diethyl-1,4,6,9-tetraphenyl- was reported for the first time in TAM as a growth promotor and an immunity enhancer [13]; however, to the best of our knowledge, there are no studies that have reported this phytochemical compound in seaweed extracts (such as TAM) prior to that of Ashour et al. [13]. This compound consists mainly of boron and silicon, which might be beneficial for plant growth because foliar applications of boron and/or silicon compounds have been found to reduce infections of powdery mildew on cucumber [32] and to improve plant growth and production [30,31]. Therefore, TAM, due to its potent bioactive compounds, was found to contain many beneficial compounds not only for plant foliar applications [57]. These identified phytochemical compounds in TAM are bioactive compounds, and they can also be found in plants. It is unclear whether foliar applications of these compounds enhance plant growth and production, but their endogenous roles in plants, such as tocopherol synthesis, are involved with phytol and the regulation of abiotic and biotic stresses by unsaturated fatty acids [62].

Seaweed extracts such as foliar spray have currently gained much beneficial significance because they can induce speedy growth and yield in cereals, vegetables, and fruit orchards, as well as in horticultural plants [4]. Several investigations have reported the different benefits of seaweed extracts on growth development and increased yield of many crops such as blackgram (*Vigna mungo*) [24], soybean (*Soybean max*) [63], wheat (*Triticum aestivum* L.) [63], tomato (*Solanum lycopersicum*) [55], rice (*Oryza sativa* L.) [1], maize (*Zea mays*) [64], and Jew's mallow (*Corchorus olitorius* L.) [4]. The findings of the present study agree with those of the many earlier studies of Layek et al. [1,65], Ashour et al. [4], and Hidangmayum and Sharma [66]. Closely related results of increased consumption of N, P, K, and Mg in cucumber (*Cucumis sativus*) with application of seaweed extracts have been previously recorded [25]. Moreover, TAM application could superiorly differ from the 100% conventional treatment. In spite of its varied ability to show a clear significant difference in all investigated measured parameters, this is interpreted accordingly to be an advantage rather than a defect. Having the same effect via application of 100% conventional mineral fertilizers could give us a promising view regarding the use of seaweed extracts as biostimulants, offering an alternative solution. If it could completely, or partially, replace the usage of chemical fertilizers, this would represent a serious addition and encourage us to save our world and environment, simultaneously aiding the conservation of biodiversity. The results of the current work were found to be well supported in the literature.

Vijayakumar et al. [67] investigated the impact of seaweed liquid (SLF) extracted from *Codium decorticatum* (at 10%, 20%, 30%, 40%, and 50% concentrations) on seed germination, yield, and quality (biochemical and pigment characteristics) of *Capsium annum* under laboratory conditions and in pots. They found that the application of SLF improved shoot

and root lengths, shoot fresh weight, and total dry weight, while the number of branches, number of pods, leaf area, and amount of chlorophyll were also increased under a 20% SLF treatment; however, the lowest values were marked at 50% SLF. On the other hand, higher concentrations of SLF were found to decrease chlorophyll content. These findings were obtained by investigations carried out on *Capsicum annum* [67], and it was noted that low concentrations of seaweed extracts have potential effects of promoting plant growth. Differently, high concentrations were also observed in wheat seedlings.

Ahmed and Shalaby [24] evaluated the production and phytochemical quality of cucumber (*Cucumis sativus*) fruits in response to the spray practice of different seaweed extracts (prepared from *Asparagopsis* spp., *Gelidium pectinutum*, and *Enteromorpha intestinalis*), a commercial seaweed extract Algreen, and compost. The results showed that the use of *E. intestinelis*, *G. pectinutum*, or commercial seaweed extracts with compost is considered an appropriate application to improve vegetative growth and yield of cucumber plants. The measured plant growth, yield components, and fruit quality parameters in our data matched the results provided by previous investigators. The length of the American cucumber ranges between 20 and 25 cm and should not be less than 15 cm [68]. The fruit's average diameter was recorded as approximately 3.17 cm with seaweed extracts; similarly, it was also found that the diameter of the cucumber varied from 5.0 to 5.7 cm and should not exceed 6.0 cm [69]. From the obtained results, it is observed that the yield of the control object was significantly lower, where foliar applications of algal extracts showed similar values to those obtained in the conventional treatment. These results are feasibly attributed to the advantages of algae when used in horticultural crops. A study on the effect of a commercial seaweed extract (Kelpak) and polyamines on nutrient-deprived (N, P, and K) okra seedlings [70] indicated that using commercial organic preparations such as Kelpak could successfully enhance the growth of nutrient-stressed crops significantly. Additionally, seaweed concentrates were capable of boosting the production of greenhouse-grown cucumbers and tepary beans under stressful conditions of a lack of nutrients. Our findings also matched those of other recently conducted studies, such as those of Valencia et al. [36], whose objective was to evaluate the production and phytochemical quality of cucumber fruits (*Cucumis sativus*) in response to the foliar application of different seaweed extracts. Evaluated treatments were *Macrocystis pyrifera*, *Bryothamnion triquetrum*, *Ascophyllum nodosum*, *Grammatophora* sp., *Macrocystis intergrifolia*, and a control treatment with inorganic fertilization. In this study, yield, quality, and phytochemical compounds of the fruit were evaluated. The study was able to determine that algal extracts increased the quality of the fruits by obtaining the highest antioxidant capacity. Thus, the use of algal extracts represents a viable option to minimize the application of conventional fertilizers, thereby reducing environmental impacts and leading to flourishing health. This equally agrees with another study carried out by investigators to report the effect of different concentrations of commercial seaweed liquid extract of *Ascophyllum nodosum* as a plant biostimulant on the growth, yield, and biochemical constituents of onions; they found the same result outputs and almost argued for similar benefits [65].

Moving forward, the results in the literature, as well as those provided by the current study, can guide recommendations for the use and exploitation of seaweed extracts and their by-products to obtain maximum advantages from all their beneficial applications.

## 5. Conclusions

Seaweed extracts are listed as environmentally friendly biofertilizers that reduce the harmful socio-economic and environmental effects of mineral fertilizers. The findings of the current study allow the conclusion that classical mineral fertilizers can be diminished to minimal concentrations by employing TAM extract. TAM is rich in phytochemical bioactive compounds, such as FAMEs, milbemycin oxime, rhodopin, nonadecane, and 5-silaspiro[4.4]nona-1,3,6,8-tetraene,3,8-bis(diethylboryl)-2,7-diethyl-1,4,6,9-tetraphenyl-. The encouraging measured parameter outcomes support the potentiality of applying TAM with and without mixes of regular NPK application. TAM could also

increase cucumber yield due to improving its chemical and physical traits related to immunity, productivity, and stress defense. Therefore, altering the overapplication of classical chemically derived N, P, and K fertilizers would extend the usage of TAM to serve organic farming. Conclusively, the application of TAM is recommended for better cucumber yield production and fruit quality.

## 6. Patents

Seaweed extract (True Algae Max (TAM)) is a patent submitted at the Egyptian Patent Office, Academy of Scientific Research and Technology (submission No. 2046/2019).

**Author Contributions:** Conceptualization, S.M.H.; G.A.G.A.G.A.G.A.; and M.A.; methodology M.A., S.M.H., and G.A.G.A.; software, M.A., G.A.G.A., and N.S.; validation, H.A.H., A.G., and L.Z.; formal analysis, S.M.H., N.S., G.A.G.A., and M.A.; investigation, G.A.G.A., A.G., and H.A.H.; resources, M.A., H.A.H., and A.G.; funding acquisition, S.M.H., M.A., A.G., and H.A.H.; data curation, H.A.H., N.S., and S.M.H.; writing—original draft preparation, S.M.H., M.A., N.S., and G.A.G.A.; writing—review and editing, M.A. and G.A.G.A.; visualization, S.M.H., M.A., H.A.H., and A.G.; supervision, S.M.H., G.A.G.A., H.A.H., and A.G.; project administration, M.A. All authors have read and agreed to the published version of the manuscript.

**Funding:** This research was funded by the Academy of Scientific Research and Technology (ASRT), Egypt, through a project titled "Prototype of sustainable marine integrated aquaculture farm for the production of seafood, valuable bio-products, and bio-diesel" (SIMAF-Project, Project ID: 1429/2016). The authors would also like to thank Taif University Researchers Supporting Project number TURSP-2020/39, Taif University, Taif, Saudi Arabia, for supporting the publication of this work.

**Institutional Review Board Statement:** Not applicable.

**Informed Consent Statement:** Not applicable.

**Data Availability Statement:** All data are provided in the manuscript.

**Acknowledgments:** The authors gratefully acknowledge Taif University Researchers Supporting Project number TURSP-2020/39, Taif University, Taif, Saudi Arabia, and Academy of Scientific Research and Technology (ASRT), Egypt, for supporting this work.

**Conflicts of Interest:** The authors declare no conflict of interest.

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
