# Peer review of "Impact of Seaweed Liquid Extract Biostimulant on Growth, Yield, and Chemical Composition of Cucumber (Cucumis sativus)"

_agriculture, doi:10.3390/agriculture11040320_

Round 1

Reviewer 1 Report

Seaweed extract biostimulants are among the supremest modern sustainable biological plant growth promoters,  eliminate plant diseases and abiotic stresses, leading to maximizing yields, and listed as environmentally friendly biofertilizersThe employment of seaweed extracts as biostimulants in agricultural practices, is very ancient. Based on their nutritional values, algal cells; either microalgae or seaweeds, represent treasury sources of dyes, proteins, fats,  polysaccharides, mineral, antioxidant, as well as plenty of biological components. They are characterized by being biodegradable, and harmless, making them environmentally friendly substances, with no chemical residuals. Seaweed extracts are highly recommended to assist organic agriculture. Thus the topic of this paper is  importance when considering the fact that studies are essential to understand the effect of different management practices on soil and plant system. In addition the topic is closely falls within the aims and scope of the journal. 
The data provided are sufficient and the statistical analysis of the results is very well presented. Tables clearly present the data. The discussion of results focus on the main points while justification of the findings are well supported by references. But some part of work is not necesery; the lines 352-371 in Discussion did not take part in presentation results of authors but presented information  in literature by different authors. 

The control  is usually object without one of factor of experiment. In this work control named C0: 100% NPK mineral fertilizer without biosomulant but more in other objects NPK. Therefore I sugest to used not the word : control but  comparing with the conventional NPK fertilization

  The Conclusions  for me is not conclusion of this work based of the aimed  and results but the futurin view of using biostimulants in buisnes and industry. It is probabli sooth and very interesting but it is not the conclusion. Therefore it should be corrected based on the achivement of work.

References
Too many, consider please if all of them are necessary.

I foud some typing mistaces:
line 129: PH -should be pH, Co3 - CO3

line 143: C. sativu-  Cucumis sativus

lines 148/150: plant -1 but per plant
line 157: 2017and 2018 -  2017 and 2018

Author Response

SUMMARY OF AUTHOR(S) RESPONSE TO REVIEWER’S COMMENTS

Manuscript #: Agriculture-1135193

Title of the Manuscript: Impact of Seaweed Liquid Extract Biostimulant on Growth, Yield, and Chemical Composition of Cucumber (Cucumis Sativus), Performed in Greenhouse

Author(s): Shimaa M. Hassan, Mohamed Ashour, Nobumitsu Sakai, Lixin Zhang, Hesham A. Hassanien, Ahmed Gaber, and Gamal Ammar

Reviewer’s Comment

Author(s) response

Reviewer  #1:

Minor Comments:

Seaweed extract biostimulants are among the supremest modern sustainable biological plant growth promoters, eliminate plant diseases and abiotic stresses, leading to maximizing yields, and listed as environmentally friendly biofertilizers. The employment of seaweed extracts as biostimulants in agricultural practices, is very ancient. Based on their nutritional values, algal cells; either microalgae or seaweeds, represent treasury sources of dyes, proteins, fats,  polysaccharides, mineral, antioxidant, as well as plenty of biological components. They are characterized by being biodegradable, and harmless, making them environmentally friendly substances, with no chemical residuals. Seaweed extracts are highly recommended to assist organic agriculture. Thus the topic of this paper is importance when considering the fact that studies are essential to understand the effect of different management practices on soil and plant system. In addition the topic is closely falls within the aims and scope of the journal.

The data provided are sufficient and the statistical analysis of the results is very well presented. Tables clearly present the data. The discussion of results focus on the main points while justifications of the findings are well supported by references.

We would like to thank “Reviewer #1” for these valuable words about our current manuscript.

But some part of work is not necesery; the lines 352-371 in Discussion did not take part in presentation results of authors but presented information in literature by different authors.

Thanks a lot for your valuable note. This should be taken into consideration, reviewed, and justified in the original manuscript as advised.

The control is usually object without one of factor of experiment. In this work control named C0: 100% NPK mineral fertilizer without biosomulant but more in other objects NPK. Therefore I suggest to used not the word : control but  comparing with the conventional NPK fertilization

Has been corrected in the whole manuscript.

  The Conclusions for me is not conclusion of this work based of the aimed and results but the future in view of using biostimulants in buisnes and industry. It is probabli sooth and very interesting but it is not the conclusion. Therefore it should be corrected based on the achievement of work.

Thanks a lot for your valuable note. The conclusion part has been corrected based on the work achievements  (Lines: 426-436).

line 129: PH -should be pH, Co3 - CO3

Has been corrected. (Table 1).

line 143: C. sativu Cucumis sativus

Has been changed. (Line: 162, 169).

lines 148/150: plant -1 but per plant

Has been corrected (Lines: 167, 170).

line 157: 2017 and 2018 -  2017 and 2018

Has been corrected (Line: 177)

References:  Too many, consider please if all of them are necessary.

Considered.

Finally, we would like to extend our sincere thanks and appreciation to the reviewers and editorial board. In fact, their comments and guidance added a lot to the research and increased its scientific content. We would also like to express our gratitude for their time and effort they put in evaluating this research.

Reviewer 2 Report

Manuscript title: Impact of seaweed liquid extract biostimulant on growth, yield, and chemical composition of cucumber (Cucumis sativus), performed in greenhouse.

 Seaweed extracts are listed as environmentally friendly biofertilizers that reduce the harmful socio-economical and environmental effects of mineral fertilizers. Additionally, they are potentially cheaper and therefore more economical alternatives to mineral fertilizers. Furthermore, in time their employment can reduce the amount of mineral fertilizers produced and applied in agriculture, with no sophisticated processes of preparation.

The aim of the study was to investigate the impact of a commercial seaweed biostimulant formally named “True Algae Max, TAM®” on growth, yield and chemical composition of cucumber (Cucumis sativus). The current experiment was performed to evaluate the impact of applying diverse levels of TAM® foliar spray, as a growth biostimulant, and NPK mineral fertilizer, alone or in combination, on the growth and yield of cucumber ‘Hesham F1’ grown under greenhouse. The experiment was performed during 2017 and 2018 seasons at Abis Experimental Farm Station, Faculty of Agriculture in Alexandria University in Egypt. The fertilizer treatments were: C0: 100% NPK mineral fertilizer as control, C25: 25% of TAM® + 75% NPK mineral fertilizer, C50: 50% of TAM® + 50% NPK mineral fertilizer, C75: 75% of TAM®+ 25% NPK mineral fertilizer, and C100: 100% of TAM®. The study showed that TAM® could increase cucumber yield, due to improving chemical and physical features of the cucumber.

The manuscript is an original research paper that follows the aim and scope of Agriculture. I would recommend deleting the phrase “performed in a greenhouse” from the title, as it is further explained in the methodology. The abstract is adequate and contains the most important information. The keywords do not repeat the words from the title, however too many keywords are used. Unnecessary keywords are: Milbemycin-oxime; 5-Silaspiro [4.4] nona-; Rhodopin; Nonadecane. The introduction clearly describes the scientific background and the current state of knowledge, but it may be supplemented with some excerpts from the Discussion chapter (e.g. line: 223-240). The material and methods are correct and were thoroughly described. The experimental design included a control sample. Statistical methods are appropriate and complete. The nomenclature is correct. The description of the results is congruent with statistical analyzes, however it is too short. The discussion is not adequate and not supported by the appropriate references to published works. 2/3 of the Discussion chapter is off topic (lines 275-351). This fragment describes the composition of the TAM and the activity of its components, which was not the subject of research. Tables 1 and 2 are found in chapter Materials and Methods, not the Results chapter. The aim of the study was to determine the impact of the biostimulant TAM on the growth and yield and chemical composition of cucumber, and not the composition and biological activity of TAM. Moreover, many of the examples given are not relevant to the research carried out (e.g. for fish, herbicidal effects, antiparasitic activity, veterinary medicine etc.).

There was no clear reference of the obtained results to the traditional mineral fertilization (control). Additionally, there are no clear conclusions on which of the fertilization variants gave the best effect. Perhaps averaging the results from two years of research (2017-2018) would facilitate inference. Such a conclusion with an indication of the optimal fertilization ratio with the use of TAM is missing in the chapter Conclusions, which is too general.

The chapter References includes as many as 89 items of literature. This is too much and needs to be reduced. The manuscript is an original research paper and is not a review article. It is necessary to choose the literature directly related to the object of the conducted experiment. The manuscript requires verification of the correctness of the English language used.

Other comments were added to the manuscript text.

Author Response

SUMMARY OF AUTHOR(S) RESPONSE TO REVIEWER’S COMMENTS

Manuscript #: Agriculture-1135193

Title of the Manuscript: Impact of Seaweed Liquid Extract Biostimulant on Growth, Yield, and Chemical Composition of Cucumber (Cucumis Sativus), Performed in Greenhouse

Author(s): Shimaa M. Hassan, Mohamed Ashour, Nobumitsu Sakai, Lixin Zhang, Hesham A. Hassanien, Ahmed Gaber, and Gamal Ammar

Reviewer’s Comment

Author(s) response

Reviewer  #2:

Minor Comments:

Seaweed extracts are listed as environmentally friendly biofertilizers that reduce the harmful socio-economical and environmental effects of mineral fertilizers. Additionally, they are potentially cheaper and therefore more economical alternatives to mineral fertilizers. Furthermore, in time their employment can reduce the amount of mineral fertilizers produced and applied in agriculture, with no sophisticated processes of preparation.

The aim of the study was to investigate the impact of a commercial seaweed biostimulant formally named “True Algae Max, TAM®” on growth, yield and chemical composition of cucumber (Cucumis sativus). The current experiment was performed to evaluate the impact of applying diverse levels of TAM® foliar spray, as a growth biostimulant, and NPK mineral fertilizer, alone or in combination, on the growth and yield of cucumber ‘Hesham F1’ grown under greenhouse. The experiment was performed during 2017 and 2018 seasons at Abis Experimental Farm Station, Faculty of Agriculture in Alexandria University in Egypt. The fertilizer treatments were: C0: 100% NPK mineral fertilizer as control, C25: 25% of TAM® + 75% NPK mineral fertilizer, C50: 50% of TAM® + 50% NPK mineral fertilizer, C75: 75% of TAM®+ 25% NPK mineral fertilizer, and C100: 100% of TAM®. The study showed that TAM® could increase cucumber yield, due to improving chemical and physical features of the cucumber.

The manuscript is an original research paper that follows the aim and scope of Agriculture.

We would like to thank “Reviewer #2” for these valuable words about our current manuscript.

I would recommend deleting the phrase “performed in a greenhouse” from the title, as it is further explained in the methodology.

Has been deleted.

The abstract is adequate and contains the most important information.

Thanks a lot for your valuable note.

The keywords do not repeat the words from the title, however too many keywords are used. Unnecessary keywords are: Milbemycin-oxime; 5-Silaspiro [4.4] nona-; Rhodopin; Nonadecane.

Has been deleted.

The introduction clearly describes the scientific background and the current state of knowledge, but it may be supplemented with some excerpts from the Discussion chapter (e.g. line: 223-240).

Has been supplemented with the suggested paragraph (Lines: 79:94).

The material and methods are correct and were thoroughly described. The experimental design included a control sample. Statistical methods are appropriate and complete. The nomenclature is correct. The description of the results is congruent with statistical analyzes, however it is too short.

Thanks a lot for your valuable note. 

The discussion is not adequate and not supported by the appropriate references to published works. 2/3 of the Discussion chapter is off topic (lines 275-351). This fragment describes the composition of the TAM and the activity of its components, which was not the subject of research.

Thanks a lot for your valuable note.  

Tables 1 and 2 are found in chapter Materials and Methods, not the Results chapter.

Fig. 1, Table 1, 2 were published before by   Ashour et al. 2020 (Ref. no. 17, Line: 110-112).

The aim of the study was to determine the impact of the biostimulant TAM on the growth and yield and chemical composition of cucumber, and not the composition and biological activity of TAM.

We reviewed the composition and biological activity of bioactive compounds of TAM in the  M&M and discussion chapters for just explaining how TAM properties could aid in achieving  the obtained results.

Many of the examples given are not relevant to the research carried out (e.g. for fish, herbicidal effects, antiparasitic activity, veterinary medicine etc.).

These topics are related to TAM biological activities, however, these topics were deleted as advised

There was no clear reference of the obtained results to the traditional mineral fertilization (control). Additionally, there are no clear conclusions on which of the fertilization variants gave the best effect. Perhaps averaging the results from two years of research (2017-2018) would facilitate inference. Such a conclusion with an indication of the optimal fertilization ratio with the use of TAM is missing in the chapter Conclusions, which is too general.

The conclusions chapter has been corrected based on the work achievements (Lines: 412-422).

The chapter References includes as many as 89 items of literature. This is too much and needs to be reduced. The manuscript is an original research paper and is not a review article. It is necessary to choose the literature directly related to the object of the conducted experiment.

Corrected.

The manuscript requires verification of the correctness of the English language used.

English language was edited and corrected.

Other comments were added to the manuscript text.

Have been corrected with track changes in the main manuscript draft.

Finally, we would like to extend our sincere thanks and appreciation to the reviewers and editorial board. In fact, their comments and guidance added a lot to the research and increased its scientific content. We would also like to express our gratitude for their time and effort they put in evaluating this research.
